# Grounding Complex Natural Language Commands for Temporal Tasks in Unseen Environments

**Jason Xinyu Liu**[*1], **Ziyi Yang**[*1], **Ifrah Idrees**[1], **Sam Liang**[2], **Benjamin Schornstein**[1],
**Stefanie Tellex**[1], **Ankit Shah**[1]
[1]Department of Computer Science, Brown University, United States
[2]Department of Computer Science, Princeton University, United States

**Abstract:** Grounding navigational commands to linear temporal logic (LTL) leverages its unambiguous semantics for reasoning about long-horizon tasks and verifying the satisfaction of temporal constraints. Existing approaches require training data from the specific environment and landmarks that will be used in natural language to understand commands in those environments. We propose Lang2LTL, a modular system and a software package that leverages large language models (LLMs) to ground temporal navigational commands to LTL specifications in environments without prior language data. We comprehensively evaluate Lang2LTL for five well-defined generalization behaviors. Lang2LTL demonstrates the state-of-the-art ability of a single model to ground navigational commands to diverse temporal specifications in 21 city-scaled environments. Finally, we demonstrate a physical robot using Lang2LTL can follow 52 semantically diverse navigational commands in two indoor environments. [1]

**Keywords:** language grounding, robot navigation, formal methods

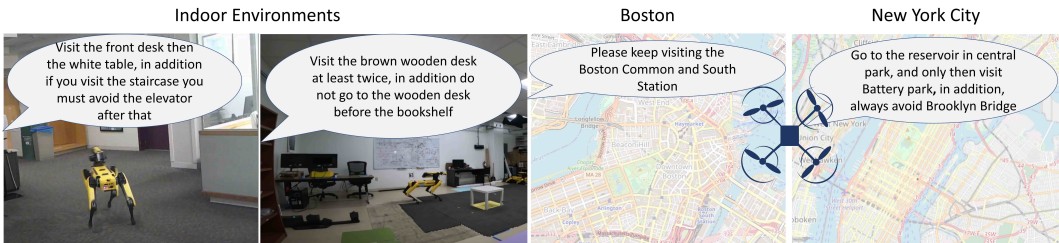

Figure 1: Lang2LTL can ground complex navigational commands in household and city-scaled environments without retraining.

## 1   Introduction

Natural language enables humans to express complex temporal tasks, like "Go to the grocery store on Main Street at least twice, but only after the bank, and always avoid the First Street under construction." Such commands contain goal specifications and temporal constraints. A robot executing this command must identify the bank as its first subgoal, followed by two separate visits to the grocery store, and visiting First Street is prohibited throughout the task execution. Linear temporal logic [1] provides an unambiguous target representation for grounding a wide variety of temporal commands.

Prior work of grounding natural language commands to LTL expressions for robotic tasks covered a limited set of LTL formulas with short lengths and required retraining for every new environment [2, 3, 4, 5]. In contrast, some recent approaches proposed leveraging LLMs to directly generate robot

---

*Equal contribution
[1]Code, datasets and videos are at https://lang2ltl.github.io/

7th Conference on Robot Learning (CoRL 2023), Atlanta, USA.

policies [6, 7, 8, 9]. The LLM output is itself a formal language that must be interpreted by an external user-defined program. Such approaches cannot solve many complex temporal tasks considered in this work. Another advantage of using a formal language like LTL is that existing work on automated planning with LTL specifications provides strong soundness guarantees.

In this paper, we propose Lang2LTL, a system capable of translating language commands to grounded LTL expressions that are compatible with a range of planning and reinforcement learning tools [10, 11, 12, 13, 14]. Lang2LTL is a modular system that separately tackles referring expression recognition, grounding these expressions to real-world landmarks, and translating a lifted version of the command to obtain the grounded LTL specification. This modular design allows Lang2LTL to transfer to novel environments without retraining, provided with a semantic map.

We formally define five generalization behaviors that a learned language grounding model must exhibit and comprehensively evaluate Lang2LTL's generalization performance on a novel dataset containing 2,125 semantically unique LTL formulas corresponding to 47 LTL formula templates. Lang2LTL showed the state-of-the-art capabilities of grounding navigational commands in 21 cities using semantic information from a map database in a zero-shot fashion. Finally, we demonstrated that a physical robot equipped with Lang2LTL was able to follow 52 semantically diverse language commands in two different indoor environments provided with semantic maps.

## 2 Preliminaries

**Large Language Models:** Autoregressive large language models (LLMs) are producing SoTA results on a variety of language-based tasks due to their general-purpose language modeling capabilities. LLMs are large-scale transformers [15] pretrained to predict the next token given a context window [16, 17]. In this work, we used the GPT series models [18, 19] and the T5-Base model [20].

**Temporal Task Specification:** Linear temporal logic (LTL) [1] has been the formalism of choice for expressing temporal tasks for a variety of applications, including planning and reinforcement learning [10, 21, 22, 11, 23, 14], specification elicitation [3, 24, 25, 26, 2], and assessment of robot actions [27]. The grammar of LTL extends propositional logic with a set of temporal operators defined as follows:

$$\varphi := \alpha \mid \neg\varphi \mid \varphi_1 \vee \varphi_2 \mid \mathbf{X}\varphi \mid \varphi_1 \mathbf{U} \varphi_2 \tag{1}$$

An LTL formula $\varphi$ is interpreted over a discrete-time trace of Boolean propositions, $\alpha \in AP$ that maps an environment state to a Boolean value. $\varphi$, $\varphi_1$, $\varphi_2$ are any valid LTL formulas. The operators $\neg$ (not) and $\vee$ (or) are identical to propositional logic operators. The temporal operator $\mathbf{X}$ (next) defines the property that $\mathbf{X}\varphi$ holds if $\varphi$ holds at the next time step. The binary operator $\mathbf{U}$ (until) specifies an ordering constraint between its two operands. The formula $\varphi_1 \mathbf{U} \varphi_2$ holds if $\varphi_1$ holds at least until $\varphi_2$ first holds, which must happen at the current or a future time. In addition, we use the following abbreviated operators, $\wedge$ (and), $\mathbf{F}$ (finally or eventually), and $\mathbf{G}$ (globally or always), that are derived from the base operators. $\mathbf{F}\varphi$ specifies that the formula $\varphi$ must hold at least once in the future, while $\mathbf{G}\varphi$ specifies that $\varphi$ must always hold.

We developed Lang2LTL based on a subset of specification patterns commonly occurring in robotics [28]. Appendix Table 4 lists the specification patterns and their interpretations.

**Planning with Temporal Task Specification:** Every LTL formula can be represented as a Büchi automaton [29, 30] thus providing sufficient memory states to track the task progress. The agent policy can be computed by any MDP planning algorithm on the product MDP of the automaton and the environment [10, 21, 22, 11, 12, 13]. Lang2LTL is compatible with any planning or reinforcement learning algorithm that accepts an LTL formula as task specification.

## 3 Related Work

**Natural Language Robotic Navigation:** Early work of language-guided robot task execution focused on using semantic parsers to ground language commands into abstract representations capable of

informing robot actions [31, 32, 33, 34, 35]. Recent work leverages large pretrained models to directly generate task plans, either as code [7, 6] or through text [36, 9]. The LLM output is a formal language that must be interpreted by an external procedure. Thus the external interpreters need to be expressive and competent for the success of these approaches. In contrast, planners for LTL offer asymptotic guarantees on the soundness of the resulting robot policies. Finally, LM-Nav [8] is a modular system that computes a navigational policy by using an LLM to parse landmarks from a command, then a vision-language model in conjunction with a graph search algorithm to plan over a pre-constructed semantic map. Note that LM-Nav can only ground commands of the *Sequence Visit* category as defined by Menghi et al. [28], while Lang2LTL can interpret 15 temporal tasks.

**Translating Language to Formal Specification:** Prior approaches are tied to the domain they were trained on and require retraining with a large amount of data to deploy in a novel domain. Gopalan et al. [3] relied on commands paired with LTL formulas in a synthetic domain called *CleanUp World*. Patel et al. [4] and Wang et al. [5] developed a weakly supervised approach that requires natural language descriptions along with a satisfying trajectory.

Leveraging LLM to facilitate translation into LTL is an emerging research direction [37, 38, 39, 40]. These approaches relied on string transformations to transform referring expressions in a command into propositions. In contrast, Lang2LTL explicitly grounds the referring expression to propositional concepts in the physical environment through a grounding module.

The closest approach to ours, proposed by Berg et al. [2], uses CopyNet [41] architecture to generate LTL formula structures followed by explicitly replacing the appropriate propositions. We demonstrate a significant performance improvement over Berg et al. [2]'s approach by leveraging LLMs as well as training and evaluating on a semantically diverse dataset.

## 4  Problem Definition

We frame the problem as users providing a natural language command $u$ to a robotic system that performs navigational tasks in an environment $\mathcal{M} = \langle \mathcal{S}, \mathcal{A}, T \rangle$, where $\mathcal{S}$ and $\mathcal{A}$ represent the states and actions of the robot, and $T$ describes the transition dynamics of the environment. Our proposed language grounding system, Lang2LTL, translates the language command $u$ to its equivalent LTL formula $\varphi$ and grounds its propositions to real-world landmarks. We assume the robot has access to its state information $\mathcal{S}$ and a semantic database of propositions $\mathcal{D} = \{k : (z, f)\}$, structured as key-value pairs. The key $k$ is a unique string identifier for each proposition, and $z$ contains semantic information about the landmark stored in a serialized format (e.g., JSON). For example, in a street map domain, the semantic information $z$ includes landmark names, street addresses, amenities, etc. $f : \mathcal{S} \rightarrow \{0, 1\}$ is a Boolean valued function that evaluates the truth value of the proposition in a given state $s$. Appendix B shows an example entry of this semantic database. Finally, we assume that the robot has access to an automated planner that accepts an LTL formula and a semantic map as input and generates a plan over the semantic map as output. We used AP-MDP [12] for this paper.

Consider the example of a drone following a given command within an urban environment depicted in Figure 2. The environment $\mathcal{M}$ encodes the position and the dynamics of the drone. The semantic database $\mathcal{D} = \{k : (z, f)\}$ includes the landmark identifiers, their semantic information, and a proximity function.

## 5  Lang2LTL: Natural Language Grounding

Lang2LTL is a modular system that leverages LLMs to solve the grounding problem by solving the following subproblems,

1. **Referring Expression Recognition**: We identify the set of substrings, $\{r_i\}$, in the command $u$ that refer to Boolean propositions. In this case, $\{r_i\}$ = {" the store on Main Street", "the bank"}.

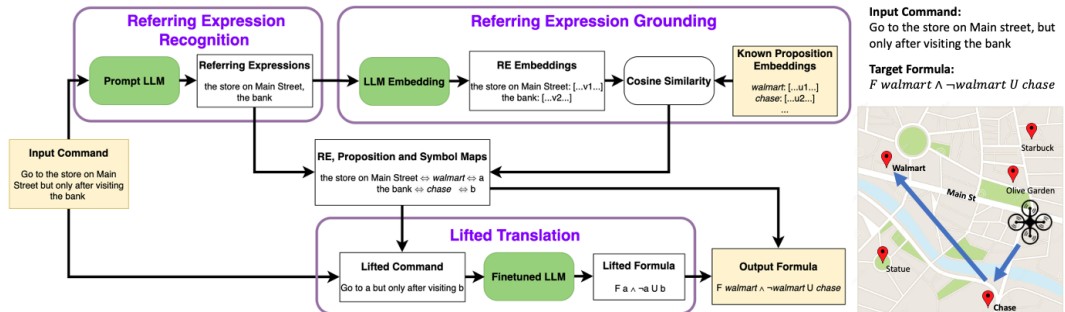

Figure 2: Lang2LTL system overview: Green blocks are pretrained or finetuned LLM models. Yellow blocks are the input or output of the system.

2. **Referring Expression Grounding**: Each referring expression $r_i$ is mapped to one of the proposition identifier strings, $k \in \mathcal{D}$, which yields a map $\{r_i \to k\}$. Each proposition is also bijectively mapped to a placeholder symbol $\beta$ from a fixed vocabulary $\boldsymbol{\beta} = \{$"A", "B", ...$\}$ by $\{k \leftrightarrow \beta\}$. In the example described above, the phrases "the store on Main Street" and "the bank" refer to the proposition identifiers $walmart$ and $chase$, which are in turn mapped to the placeholder symbols "A" and "B".

3. **Lifted Translation**: After substituting the referring expressions $\{r_i\}$ by placeholder symbols $\boldsymbol{\beta}$ using the map $\{r_i \to \beta\}$, we obtain the lifted utterance, "Go to A, but only after visiting B.". The lifted translation module then translates this lifted utterance into a lifted LTL formula, $\varphi_{\boldsymbol{\beta}} = \mathcal{F}(A) \wedge (\neg A \, \mathcal{U} \, B)$.

Finally, we substitute placeholder symbols "A" and "B" by grounded propositions $walmart$ and $chase$ using the bijection $\{k \leftrightarrow \beta\}$ to construct the output LTL formula depicted in Figure 2.

We hypothesized that the benefit of solving the translation problem in the lifted domain is two-fold. First, the output vocabulary size is significantly reduced. Secondly, the lifted formula data can be sourced from multiple domains, thus providing access to a larger training dataset. Once accurately translated in the lifted domain, the formulas can be grounded to unseen landmarks in novel domains, a task at which LLMs excel. We examine the efficacy of this modularization in Section 6.

## 5.1 Referring Expression Recognition (RER)

Referring expressions are noun phrases, pronouns, and proper names that refer to some individual objects [42]. In this work, we only consider the task of recognizing noun phrases and proper names. Referring expressions are entire substrings that refer to a single entity, therefore, they are a superset of named entities. For example, "the store on Main Street" is the referring expression, but it contains two named entities, "store" and "Main Street."

Referring expression recognition is generally challenging to all existing pretrained name entity recognition models, especially without adequate examples for finetuning. We demonstrate high performance on the RER task by adapting the GPT-4 prompted with a task description and examples to enable in-context learning. Details of the prompting approach are provided in Appendix D.

## 5.2 Referring Expression Grounding (REG)

Due to the diversity of natural language, a user can refer to a landmark using many possible referring expressions. Grounding these expressions into the correct propositions is challenging. We propose using the embeddings computed by an LLM for measuring similarity. LLMs have been shown to map semantically similar texts to similar embedding values [43].

Let $g_{embed} : r \to \mathbb{R}^n$ represent the function that computes an $n$-dimensional embedding of a text string using the parameters of the LLM. Following Berg et al. [44], we match the referring expressions $\{r_i\}$ to the proposition tokens $k$'s by matching their respective embeddings using cosine similarity.

The embedding of a proposition token, $k$, is computed by encoding the semantic information, $z$, of its corresponding landmark, e.g., street name and amenity. This process is represented formally as follows,

$$k^* = \underset{\{k:(z,f)\} \in \mathcal{D}}{\operatorname{argmax}} \frac{g_{embed}(r_i)^\top g_{embed}(z)}{||g_{embed}(r_i)|| \, ||g_{embed}(z)||} \tag{2}$$

## 5.3 Lifted Translation

Our lifted translation module operates with a much smaller vocabulary than the number of landmarks within any given navigation domain. It can also be trained with navigational commands from a wider variety of data sources. In designing our lifted translation module, we followed the Gopalan et al. [3] and Patel et al. [4] and represented the prediction target LTL formulas in the prefix format instead of the infix format. This allows us to unambiguously parse the formulas without requiring parenthesis matching and shorten the output.

The lifted translation module accepts a lifted utterance as an input and generates a lifted LTL formula with an output vocabulary of up to 10 operators and five lifted propositions. We evaluated the following model classes for lifted translation. The implementation details are provided in Appendix E.

**Finetuned LLM:** We finetuned LLMs using supervised learning following [16]. We tested two LLMs with supervised finetuning, namely, T5-Base (220M) [20] (using the Hugging Face Transformer library [45], and the text-davinvi-003 version of GPT-3 using the OpenAI API. The target was an exact token-wise match of the ground-truth LTL formula.

**Prompt LLM:** We evaluated prompting the pre-trained GPT-3 [18] and GPT-4 [19] models using the OpenAI API. We did not vary the prompts throughout a given test set.

**Seq2Seq Transformers:** We trained an encoder-decoder model based on the transformer architecture [15] to optimize the per-token cross entropy loss with respect to the ground-truth LTL formula, together with a token embedding layer to transform the sequence of input tokens into a sequence of high-dimensional vectors.

## 6  Evaluation of Language Grounding

We tested the performance of Lang2LTL towards five definitions of generalizing temporal command interpretation as informed by formal methods in Section 6.1. We evaluated the performance of each module described in Section 5 in isolation in addition to a demonstration of the integrated system. Lang2LTL achieved state-of-the-art performance in grounding diverse temporal commands in 21 novel *OpenStreetMap* regions [46].

### 6.1  Generalization in Temporal Command Understanding

Utilizing a formal language, such as LTL, to encode temporal task specifications allows us to formally define five types of generalizing behaviors.

**Robustness to Paraphrasing:** Consider two utterances, $u_1 =$ " Go to $chase$", and $u_2 =$ "Visit $chase$," describing the same temporal formula $\mathbf{F}\ chase$. If the system has only seen $u_1$ at training time, but it correctly interprets $u_2$ at test time, it is said to demonstrate robustness to paraphrasing. This is the most common test of generalization we observe in prior works on following language commands for robotics. A test-train split of the dataset is adequate for testing robustness to paraphrasing, and we refer to such test sets as *utterance holdout*. Most prior works [3, 2, 40, 37, 5, 4] demonstrate robustness to paraphrasing.

**Robustness to Substitutions:** Assume the system has been trained to interpret the command corresponding to $\mathbf{F}\ chase$, and $\mathbf{G}\ \neg walmart$ at training time but has not been trained on a command corresponding to $\mathbf{F}walmart$. If the system correctly interprets the command corresponding to

**F**$walmart$, it is said to demonstrate robustness to substitution. To test for robustness to substitutions, any formula in the test set must not have a semantically equivalent formula in the training set. [3] demonstrated limited robustness to substitutions. The lifted translation approach followed by Berg et al. [2], NL2TL [40], and Lang2LTL demonstrates robustness to substitutions

**Robustness to Vocabulary Shift**: Assume the system has been trained to interpret commands corresponding to **F**$chase$ at training time but has never seen any command containing the propositions $walmart$. If the system correctly interprets **F**$walmart$ at test time, the system is said to be robust to vocabulary shift. To test for robustness to vocabulary shift, we identify the set of unique propositions occurring in every formula in the test and training set. The training set and test set vocabularies should have an empty intersection in addition to the non-equivalence of every test formula with respect to the training formula. Methods that explicitly rely on lifted representations show robustness to vocabulary shifts by design [44, 8, 40]. Our full system evaluations demonstrated robustness to novel vocabularies in Section 6.5.

**Robustness to Unseen Formulas**: Assume that all formulas in the training set are transformed by substituting the propositions in a pre-defined canonical order, e.g., both **F** $chase$, and **F** $walmart$ are transformed to **F** $a$. We refer to these transformed formulas as the formula skeleton. To test for robustness to unseen formulas, the test set must not share any semantically equivalent formula skeleton with the training set. We used the built-in equivalency checker from the Spot LTL library [47].

**Robustness to Unseen Template Instances**: We define this as a special case of robustness to unseen formulas. If all the formulas in the test and training set are derived from a template-based generator using a library of pre-defined templates [28, 48], a model may exhibit generalization to different semantically distinct formula skeletons of the same template; such a model displays robustness to unseen templates. We refer to the test set whose skeletons vary only as instantiations of templates seen during training as a *formula holdout* test set. If the unseen formulas in the test set do not correspond to any patterns, we refer to it as a *type holdout* test set.

None of the prior works have evaluated proposed models for robustness to unseen formulas. We evaluated the lifted translation module of Lang2LTL on both *formula* and *type holdout* test sets. Lang2LTL experienced a degradation of performance as expected, indicating that robustness to unseen formulas is still an open challenge.

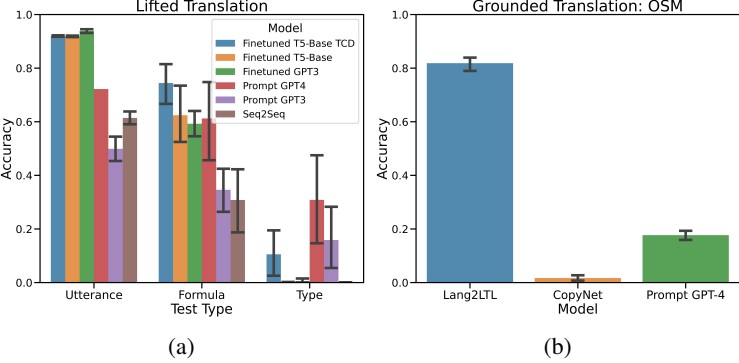

Figure 3: Figure 3a depicts the average accuracies of six lifted translation models on the three holdout sets over five-fold cross-validation. Figure 3b depicts the average accuracies of the grounded translation task in the *OSM* domain.

Table 1: Proposition Grounding Evaluation

| Component | Accuracy |
|---|---|
| RE Recognition | $98.01 \pm 2.08\%$ |
| RE Grounding | $98.20 \pm 2.30\%$ |

Table 2: Cross-Domain Evaluation. * for Zero-Shot

| | *OSM* [44] | *CleanUp* [3] |
|---|---|---|
| Lang2LTL | $49.40 \pm 15.49\%^*$ | $78.28 \pm 1.73\%^*$ |
| CopyNet [44] | $45.91 \pm 12.70\%$ | $2.57\%^*$ |
| RNN-Attn [3] | NA$^*$ | $95.51 \pm 0.11\%$ |

## 6.2 Lifted Dataset

We first collected a new parallel corpus of 1,156 natural language commands in English and LTL formulas with propositional symbols to train and evaluate the lifted translation module. We included 15 LTL templates identified by Menghi et al. [28] and generated 47 unique lifted LTL formula templates by varying the number of propositions from 1 to 5 when applicable.

To improve the lifted translation module's *Robustness to Substitutions*, we permuted the propositions in utterances and corresponding formulas. The lifted dataset after permutation contains 2,125 unique LTL formulas and 49,655 English utterances describing them. For a detailed description of the dataset, please refer to Appendix H.

## 6.3 Grounded Dataset

**Generating Diverse REs:** We used GPT-4 to paraphrase landmark names from an open-source map database, *OpenStreetMap* (*OSM*) [46], to more diverse forms of referring expressions (REs) by providing the semantic information. The prompt used for generating diverse REs is in Appendix C.

We then substituted the symbols in the lifted dataset (Section 6.2) by diverse REs on 100 randomly sampled lifted utterances for each of the 21 *OSM* cities. For a list of example commands, please see Appendix Table 5.

## 6.4 Component-wise Evaluation

**Proposition Grounding:** We evaluated the RER and REG modules on the grounded *OSM* dataset with 2,100 utterances across 21 cities. The average accuracy and the standard error across the cities for these modules are depicted in Table 1. The accuracy of RER decreased slightly, and REG performed uniformly well as we varied the complexity of commands and REs, respectively (Appendix Figure 6).

**Lifted Translation:** We evaluated the six models presented in Section 5.3 for lifted translation through five-fold cross-validation on *utterance*, *formula*, and *type holdout* tests. Figure 3a depicts the average accuracy and the standard deviation across the folds. We note that the two finetuned LLM models demonstrate the best performance on utterance holdout. We also noted a degradation of performance on *formula* and *type holdout* tests, with the latter being the most challenging across all models. Finetuned LLM models suffered the worst degradation of performance. Finally, the addition of type-constrained decoding to T5 significantly improved its performance on the more challenging *formula* and *type holdout*. Due to the lower cost of inference, and the ability to implement type-constrained decoding to prevent syntax errors, we chose the finetuned T5 model for lifted translation in our full-system evaluation.

## 6.5 System Evaluation

We compared the translation accuracy of the full Lang2LTL system on the grounded datasets with the CopyNet-based translation model [2] and Prompt GPT-4. We retrained CopyNet with an identical data budget as the Lang2LTL lifted translation model. For Prompt GPT-4, we ensured that there was at least one example from each formula skeleton in the dataset included in the prompt. Figure 3b depicts Lang2LTL outperforming both the baselines by a significant margin. Note that due to the high cost of inference, Prompt GPT-4 was only evaluated on a smaller subset of the test set.

## 6.6 Cross-Domain Evaluation

We further tested the zero-shot generalization capability of Lang2LTL on two different crowd-sourced datasets from prior work; the Cleanup World [3] on an analog indoor environment; and the *OSM* dataset [2] collected via Amazon Mechanical Turk. Table 2 shows the translation accuracies of Lang2LTL without any fine-tuning on the target datasets. Note that Lang2LTL outperforms CopyNet results reported by Berg et al. [2]. We further note that the CleanUp World dataset contains 6 unique

formula skeletons, out of which some were not a part of our lifted dataset. The degraded performance is expected when the model needs to generalize to unseen formulas.

## 7 Robot Demonstration

To demonstrate Lang2LTL's ability to directly execute language commands by interfacing with automated planning algorithms, we deployed a quadruped module robot, Spot [49] with the AP-MDP planner [12] in two novel indoor environments. Each environment had eight landmarks (e.g., bookshelf, desks, couches, elevators, tables, etc.). We ensured that each environment had multiple objects of the same type but different attributes. We used Spot's GraphNav framework to compute a semantic map of the environment. The AP-MDP [12] planner can directly plan over this semantic map, given an LTL task specification. Please refer to Appendix J for a complete description of the task environments.

As a proof-of-concept, we further finetuned the lifted translation module on 120,000 lifted utterances and formulas formed by sampling pairwise compositions from the lifted database and composed using conjunctions and disjunctions.

We compared Lang2LTL to Code-as-Policies (CaP) [6], a prominent example of directly grounding language instructions to robot plans expressed as Python code. We provided Code-as-Policies with interface functions to input the semantic map and a helper function to automatically navigate between nodes while having the ability to avoid certain regions. Thus CaP had access to actions at a much higher level of abstraction than our system. Note that AP-MDP only interfaced with primitive actions defined as movement between any two neighboring nodes.

Lang2LTL was able to correctly ground 52 commands (40 satisfiable and 12 unsatisfiable commands). The failure cases were due to incorrect lifted translations. The formal guarantees of the AP-MDP planner assured that the robot execution was aborted when facing an unsatisfiable specification. By contrast, CaP was only able to generate acceptable executions for 23 out of 52 commands and did not explicitly recognize the unsatisfiable commands. CaP demonstrated more robustness to paraphrasing than our system, which failed on some compositional patterns not in the augmented lifted dataset.

## 8 Limitations

We observed that Lang2LTL fails at grounding language commands with certain utterance structures, which suggests that the lifted translation model overfits some training utterances. A list of incorrect groundings is shown in Appendix Table 2 and Table 3. Finetuning models with larger capacities, e.g., T5-Large, may help. In this work, we only consider the task of recognizing noun phrases and proper names as referring expressions, not pronouns. We can tackle the coreference resolution problem by first prompting an LLM or using an off-the-shelf model to map pronouns to their corresponding noun phrases or proper names before the RER module. If there are multiple landmarks of the same semantic features present in the environment, e.g., two Starbucks, Lang2LTL cannot distinguish the two and selects one at random. To resolve this ambiguity, the robot needs to actively query the human user via dialog when necessary.

## 9 Conclusion

We propose Lang2LTL, a modular system using large language models to ground complex navigational commands for temporal tasks in novel environments of household and city scale without retraining and generalization tests for language grounding systems. Lang2LTL achieves a grounding accuracy of $81.83\%$ in 21 unseen cities and outperforms the previous SoTA and an end-to-end prompt GPT-4 baseline. Any robotic system equipped with position-tracking capability and a semantic map with landmarks labeled with free-form text can utilize Lang2LTL to interpret natural language from human users without additional training.

**Acknowledgments**

The authors would like to thank Charles Lovering for his feedback that helped improve the draft, Peilin Yu and Mingxi Jia for helping edit the videos, and Alyssa Sun for developing the web demonstration. This work is supported by ONR under grant numbers N00014-22-1-2592 and N00014-21-1-2584, AFOSR under grant number FA9550-21-1-0214, NSF under grant number CNS-2038897, and with support from Echo Labs and Amazon Robotics.

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

## Appendix A    Specification Patterns

We developed Lang2LTL to ground navigational commands to LTL formulas. We started with the catalog of robotic mission-relevant LTL patterns for robotic missions by Menghi et al. [28]. We adopted 15 templates that are relevant to robot navigation and modified some of their patterns to semantically match our requirements. The complete list of pattern descriptions and the corresponding LTL templates is in Table 4. Note that we use some additional abbreviated temporal operators, specifically "Weak until" $\mathbf{W}$, and the "Strong release" $\mathbf{M}$ in terms of standard operators, i.e., $a \mathbf{W} b = a \mathbf{U} (b \vee \mathbf{G}a)$, and $a \mathbf{M} b = b \mathbf{U} (a \wedge b)$.

## Appendix B    Semantic Information from OpenStreetMap Database

We show an example entry of the semantic dataset as follows,

```
{
    "Jiaho supermarket": {
        "addr:housenumber": "692",
        "shop": "supermarket",
        "opening_hours": "Mo-Su 08:00-20:00",
        "phone": "6173389788",
        "addr:postcode": "02111",
        "addr:street": "Washington Street"
    },
    ...,
}
```

## Appendix C    Implementation Details about Referring Expression Generation

We prompt the GPT-4 model for paraphrasing landmarks with corresponding OSM databases. For each landmark, three referring expressions are generated. The prompt for generating referring expressions by paraphrasing is as follows,

Use natural language to describe the landmark provided in a python dictionary form in a short phrase.

Landmark dictionary:
'Fortuna Cafe': {'addr:housenumber': '711', 'cuisine': 'chinese', 'amenity': 'restaurant', 'addr:city': 'Seattle', 'addr:postcode': '98104', 'source': 'King County GIS;data.seattle.gov', 'addr:street': 'South King Street'}
Natural language:
Chinese cafe on South King Street

Landmark dictionary:
'Seoul Tofu House & Korean BBQ': {'addr:housenumber': '516', 'cuisine': 'korean', 'amenity': 'restaurant', 'addr:city': 'Seattle', 'addr:postcode': '98104', 'source': 'King County GIS;data.seattle.gov', 'addr:street': '6th Avenue South'}
Natural language:
Seoul Tofu House

Landmark dictionary:
'AI Video': {'shop': 'electronics'}
Natural language:
AI Video selling electronics

...

Landmark dictionary:
'Dochi': {'addr:housenumber': '604', 'cuisine': 'donut', 'amenity': 'cafe'}

Natural language:
a cafe selling donut named Dochi

Landmark dictionary:
'AVA Theater District': {'addr:housenumber': '45', 'building': 'residential', 'building:levels': '30'}
Natural language:
AVA residential building

Landmark dictionary:
'HI Boston': {'operator': 'Hosteling International', 'smoking': 'no', 'wheelchair': 'yes', 'tourism': 'hostel'}
Natural language:
HI Boston

Landmark dictionary:

## Appendix D   Implementation Details about Referring Expression Recognition

The prompt for referring expression recognition is as follows,

> Your task is to repeat exact strings from the given utterance which possibly refer to certain propositions.
>
> Utterance: move to red room
> Propositions: red room
>
> Utterance: visit Cutler Majestic Theater
> Propositions: Cutler Majestic Theater
>
> Utterance: robot c move to big red room and then move to green area
> Propositions: big red room | green area
>
> Utterance: you have to visit Panera Bread on Beacon Street, four or more than four times
> Propositions: Panera Bread on Beacon Street
>
> Utterance: go to Cutler Majestic Theater at Emerson College on Tremont Street, exactly three times
> Propositions: Cutler Majestic Theater at Emerson College on Tremont Street
>
> ...
>
> Utterance: make sure you never visit St. James Church, a Christian place of worship on Harrison Avenue, Dunkin' Donuts, Thai restaurant Montien, New Saigon Sandwich, or Stuart St @ Tremont St
> Propositions: St. James Church, a Christian place of worship on Harrison Avenue | Dunkin' Donuts | Thai restaurant Montien | New Saigon Sandwich | Stuart St @ Tremont St
>
> Utterance: move the robot through yellow region or small red room and then to large green room
> Propositions: yellow region | small red room | large green room
>
> Utterance:

The results of using this prompt to recognize referring expressions with spatial relations are shown in Table 6.

## Appendix E    Implementation Details about Lifted Translation

### E.1    Fintuned T5-Base

For finetuning the T5-Base model, we set the batch size to 40, the learning rate to $10^{-4}$, and the weight decay to $10^{-2}$. We ran training for 10 epochs and picked the best-performing one for reporting results.

### E.2    Finetuned GPT-3

The per specification type accuracies and the accuracies for varying number of propositions in the formula while testing the finetuned GPT-3 model on the utterance holdout is depicted in Figure 3. The finetuned GPT-3 model achieves high accuracies across formula types and varying numbers of propositions. It shows the benefit of having a large high-quality dataset of natural language commands representing diverse LTL formulas. All previous works also used utterance holdout as their testing methodology, but their training and test sets contain significantly fewer unique LTL formulas.

### E.3 Prompt GPT-4

The prompt for end-to-end GPT-4 is as follows,

> Your task is to translate English utterances into linear temporal logic (LTL) formulas.
>
> Utterance: visit b
> LTL: F b
>
> Utterance: eventually reach b and h
> LTL: & F b F h
>
> Utterance: go to h a and b
>
> LTL: & F h & F a F b
>
> Utterance: proceed to reach h at the next time instant only and only if you see b
> LTL: G e b X h
>
> Utterance: wait at b till you see h
> LTL: U b h
>
> Utterance: go to h in the very next time instant whenever you see b
> LTL: G i b X h
>
> Utterance:

### E.4 Prompt GPT-3

The prompt for end-to-end GPT-3 is the same as the one we used for Prompt GPT-4.

### E.5 Seq2Seq Transformer

We constructed and trained a transformer model following [15]. More specifically, we built the model's encoder with three attention layers and decoder with three layers, and we used 512 as the embedding size and 8 as the number of attention heads. For training, we adapted batched training with a batch size equal to 128, learning rate equal to $10^{-4}$, and dropout ratio equal to 0.1; the training process runs for 10 epochs, and we picked the best-performing checkpoint for baseline comparison.

### E.6 Type Constrained Decoding (TCD)

Constrained decoding has been used in generating formal specifications for eliminating syntactically invalid outputs. Due to the sampling nature of NN-based models, generated tokens from the output layer can result in syntactical errors that can be detected on the fly, and type-constrained decoding solves it by forcing the model to only generate tokens following the correct grammar rule. By eliminating syntax errors, it also improves the overall performance of the system.

In practice, type-constrained decoding is implemented at each step of the decoding loop: first checking the validity of the output token, then appending the valid token or masking the invalid, and re-generating a new token according to the probability distribution after masking. In addition, we design an algorithm to simultaneously enforce the length limitation and syntactical rule by parsing partial formulas into binary trees. Beyond a given maximum height of the tree, the model is forced only to generate propositions but not operators.

## Appendix F    Implementation Details about Code as Policies

We designed two prompts for reproducing Code as Policies: one for code generation and the other for parsing landmarks. The code generation prompt is expected to generate an executable Python script that calls the goto_loc() function for traversing through the environment and psrse_loc() function to ground referring expressions to landmarks, where the landmark resolution prompt is used. The code generation prompt for graph search is as follows,

```python
# Python 2D robot navigation script

import random
from utils import goto_loc, parse_loc

# make the robot go to wooden desk.
target_loc = parse_loc('wooden desk')
goto_loc(target_loc)
# go to brown desk and then white desk.
target_loc_1 = parse_loc('brown desk')
target_loc_2 = parse_loc('white desk')
target_locs = [target_loc_1, target_loc_2]
for target_loc in target_locs:
    goto_loc(target_loc)
# head to doorway, but visit white kitchen counter before that.
target_loc_1 = parse_loc('white kitchen counter')
target_loc_2 = parse_loc('doorway')
target_locs = [target_loc_1, target_loc_2]
for target_loc in target_locs:
    goto_loc(target_loc)
# avoid white table while going to grey door.
target_loc = parse_loc('grey door')
avoid_loc = parse_loc('white table')
target_locs = [target_loc]
avoid_locs = [avoid_loc]
for loc in target_locs:
    goto_loc(loc, avoid_locs=avoid_locs)
# either go to steel gate or doorway
target_loc_1 = parse_loc('steel gate')
target_loc_2 = parse_loc('doorway')
target_locs = [target_loc_1, target_loc_2]
target_loc = random.choice(target_locs)
goto_loc(target_loc)

...

# go to doorway three times
target_loc = parse_loc('doorway')
for _ in range(3):
    goto_loc(target_loc)
    random_loc = target_loc
    while random_loc == target_loc:
        random_loc = random.choice(locations)
    goto_loc(random_loc)
```

The landmark resolution prompt is as follows,

```
# Python parsing phrases to locations script

locations = ['bookshelf', 'desk A', 'table', 'desk B', 'doorway', 'kitchen counter', 'couch', 'door']
semantic_info = {
  "bookshelf": {"material": "wood", "color": "brown"},
  "desk A": {"material": "wood", "color": "brown"},
  "desk B": {"material": "metal", 'color': "white"},
  "doorway": {},
  "kitchen counter": {"color": "white"},
  "couch": {"color": "blue", "brand": "IKEA"},
  "door": {"material": "steel", "color": "grey"},
  "table": {"color": "white"},
  }
# wooden brown bookshelf
ret_val = 'bookshelf'

...

locations = ['bookshelf', 'desk A', 'table', 'desk B', 'doorway', 'kitchen counter', 'couch', 'door']
semantic_info = {
  "bookshelf": {"material": "wood", "color": "brown"},
  "desk A": {"material": "wood", "color": "brown"},
  "desk B": {"material": "metal", 'color': "white"},
  "doorway": {},
  "kitchen counter": {"color": "white"},
  "couch": {"color": "blue", "brand": "IKEA"},
  "door": {"material": "steel", "color": "grey"},
  "table": {"color": "white"},
  }
# blue IKEA couch
ret_val = 'couch'
```

## Appendix G   Implementation Details about Grounded Translation

### G.1   CopyNet

For reproducing [44], we trained the CopyNet baseline with our grounded dataset preprocessed as its required format. To make a fair comparison on generalization ability, the CopyNet model has only seen utterance-formula pairs from the Boston subset, and the evaluation is run on grounded datasets of the rest 21 cities. For training CopyNet, we followed closely the instructions of the original paper and used the exact same LSTM model structure and pre-computed glove embedding for landmark resolution. On the hyperparameters, we set the embedding size to 128, the hidden size to 256, the learning rate to $10^{-3}$, and the batch size to 100.

### G.2   Prompt GPT-4

The prompt for end-to-end GPT-4 is as follows. While we tried including a landmark list in the prompt, it was removed in the final version because we observed empirically that Prompt GPT-

Table 1: Dataset Comparison

|  | Lang2LTL Lifted | CleanUp World | NL2TL | Wang et al. [5] |
|---|---|---|---|---|
| Number of datapoints | 49,655 | 3,382 | 39,367 | 6,556 |
| Unique formula skeletons | 47 | 4 | 605 | 45 |
| #Propositions (min, max, mean) | (1, 5, 3.79) | (1, 3, 1.85) | (1, 7, 2.86) | (1, 4, 2.01) |
| Formula Length (min, max, mean) | (2, 67, 18.89) | (2, 7, 4.77) | (1, 13, 5.98) | (3, 7, 4.48) |

4 achieved better performance without explicitly giving a list of landmarks during prompt engineering.

Your task is to first find referred landmarks from a given list then use them as propositions to translate English utterances to linear temporal logic (LTL) formulas.

Utterance: visit Panera Bread sandwich fast food on Stuart Street
LTL: F panera_bread

Utterance: eventually reach Wang Theater, and The Kensington apartments
LTL: & F wang_theater F the_kensington

...

Utterance: make sure that you have exactly three separate visits to Seybolt Park
LTL: M & seybolt_park F & ! seybolt_park F & seybolt_park F & ! seybolt_park F seybolt_park | ! seybolt_park G | seybolt_park G | ! seybolt_park G | seybolt_park G | ! seybolt_park G | seybolt_park G ! seybolt_park

Utterance:

## Appendix H Dataset Details

### H.1 Quantifying diversity of temporal commands

We quantify the diversity of the temporal commands a system is tested on using the temporal formula skeletons in the evaluation corpus of commands. We propose that each novel dataset should be characterized along the following dimensions, and as an example, we provide the respective values for the Lang2LTL dataset (lifted and grounded OSM dataset) described in Section 6.4 of the main paper.

1. Number of semantically unique formulas: 47
2. Number of propositions per formula: minimum: 1, maximum: 5, average: 3.79
3. Length of formulas: minimum: 2, maximum: 67. average: 18.89
4. Vocabulary size (for grounded datasets): 1757
5. Linguistic diversity of utterances: self-BLEU score: 0.85

Table 1 compares our proposed lifted dataset and other datasets proposed in prior work.

## Appendix I Detailed Result Analysis on Lifted Translation

We further analyzed the results for each model and holdout type for the lifted translation problem. In particular, we computed the accuracies per each formula type and the number of unique propositions required to construct the target formula. This analysis provides insights into the sensitivity of the models to particular templates and formula lengths.

The accuracies of each model and holdout type categorized by formula types are depicted in Figure 1. We observe that for both the finetuned models (Finetuned T5 and Finetuned GPT-3), the model

achieves high accuracies across various formula types for *Utterance Holdout*. Note that the performance across types is more uniform for the Finetuned GPT-3 than the Finetuned T5-Base model. Next, we note that Prompt GPT-4 achieves better accuracies as compared to Prompt GPT-3 across all evaluations.

We observe that the performance of the finetuned models is more unbalanced across different formula types for the *Formula Holdout* test case. In comparison, Prompt GPT models achieve non-zero accuracies across all formula types. Once again, Prompt GPT-4 outperforms Prompt GPT-3. We note that adding type-constrained decoding to Finetuned T5-Base during inference only marginally improved *Utterance Holdout*, but significantly improved *Formula* and *Type Holdout*, which implies Finetuned T5-Base model is more likely to produce syntactically incorrect output when the grounding formula instance or type have not seen during training.

Finally, we note that only the prompt GPT models achieve meaningful accuracies in the *Type Holdout* scenarios. However, even in *Type Holdout*, the accuracies are concentrated on formula types that only had short lengths or shared subformulas with types seen during training. We can conclude that *Formula* and *Type Holdout* remain challenging paradigms of generation and an open problem for automated translation of language commands into formal specifications.

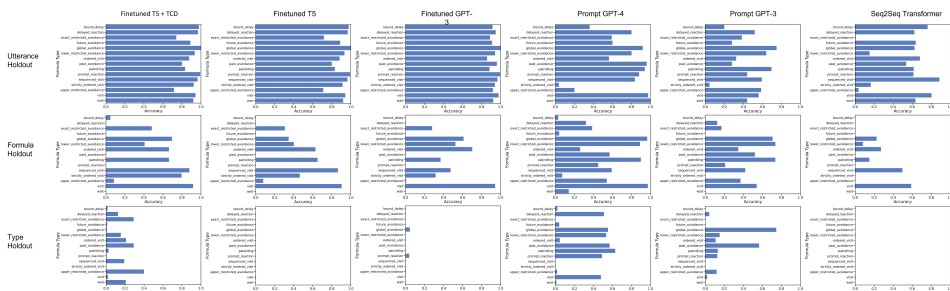

Figure 1: The accuracies per grounding formula types of six lifted translation models

Next, we repeated the above analysis but categorized accuracies by the number of unique propositions that appear within a formula. The results are depicted in Figure 2.

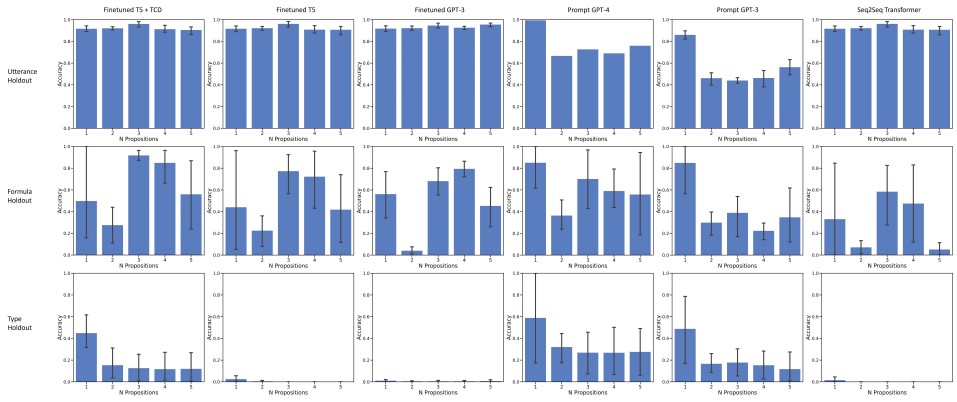

Figure 2: The accuracies per number of unique propositions of six lifted translation models

Here we note that in the *Utterance Holdout* test, the finetuned models demonstrated balanced performance across the dataset, whereas both prompt GPT models demonstrated degraded performance when the number of unique propositions in the formula was increasing. Subsequently, the degraded performance on longer formulas was apparent even within the *Formula* and *Type holdout* domains. In contrast, the three finetuned models performed better for longer formulas in *Formula Holdout*. We hypothesize that this is because the finetuned models were more able to generalize to different formula lengths of the same template (in particular, the templates that required temporal ordering

constraints to be encoded) as compared to the prompt completion-based approaches. In addition, there are more samples in the training set for longer formulas due to permutations of propositions.

As finetuning an LLM on the target task produced the best results for *Utterance Holdout*, we further analyzed the cause of errors for the instances where the lifted translation was incorrect. We categorize the errors as follows:

1. **Syntax Errors:** The formula returned by the lifted translation module was not a valid LTL formula.

2. **Misclassifed formula type:** The lifted translation module returns an identifiable but incorrect formula type that did not correspond to the input command.

3. **Incorrect propositions:** The returned formula was of the correct formula type but had the incorrect number of propositions.

4. **Incorrect permutation:** The formula was of the correct template class and had the right number of propositions, but the propositions were in the wrong location within the formula.

5. **Unknown template** The returned formula was a valid LTL formula but did not belong to any known formula types.

Figure 3 to Figure 5 depict the relative frequencies of the error cases as a pie chart for the three finetuned models. Note that returning unknown formula templates with the correct syntax was the most common cause of error in the lifted translation based on all finetuned models.



Figure 3: Error frequencies of Finetuned T5-Base with TCD

Figure 4: Error frequencies of Finetuned T5-Base

Figure 5: Error frequencies of Finetuned GPT-3

Since Finetuned GPT-3 achieves the best generalization across formula types, and type-constrained decoding (TCD) during inference significantly improves the translation accuracies for unseen formula instances and types, the combination of large language models and TCD is by far the best approach for grounding language commands for temporal tasks.

## Appendix J  Robot Demonstration

### J.1  Indoor Environment #1

The semantic information of landmarks in the first household environment is as follows,

```
{
    "bookshelf": {
        "material": "wood",
        "color": "brown"
    },
    "desk A":{
        "material": "wood",
        "color": "brown"
    },
    "desk B": {
        "material": "metal",
        "color": "white"
    },
```

```
    "doorway": {},
    "kitchen counter": {
        "color": "white"
    },
    "couch": {
        "color": "blue",
        "brand": "IKEA"
    },
    "door": {
        "material": "steel",
        "color": "grey"
    },
    "table": {
        "color": "white"
    }
}
```

Natural language commands used to test our system Lang2LTL and Code as Polices [6] are shown in Table 2.

## J.2 Indoor Environment #2

The semantic information of landmarks in the second household environment is as follows,

```
{
    "hallway A": {
        "decoration": "painting"
    },
    "hallway B": {
        "decoration": "none"
    },
    "table A": {
        "location": "kitchen",
        "material": "metal",
        "color": "blue"
    },
    "table B": {
        "location": "atrium",
        "material": "metal",
        "color": "white"
    },
    "classroom": {
        "door": ["glass", "grey"]
    },
    "elevator": {
        "color": "purple"
    },
    "staircase": {},
    "front desk": {},
    "office": {
        "door": ["wood", "yellow"]
    },
}
```

Natural language commands used to test our system Lang2LTL and Code as Polices [6] are shown in Table 3.

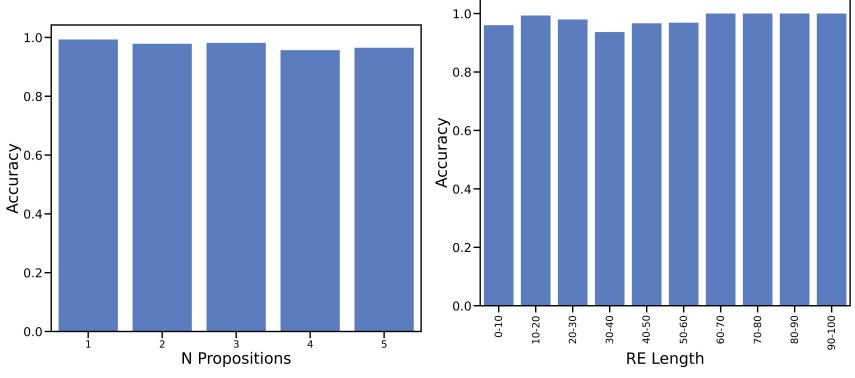

(a) RER Accuracy vs. Command Complexity    (b) REG Accuracy vs. RE Complexity

Figure 6: Figure 6a shows the accuracies of the referring expression recognition (RER) module as the complexity of commands (measured by the number of referring expressions in the command) increases. Figure 6b shows the accuracy of the referring expression grounding (REG) module as the complexity of REs (measured by string length) increases.

Table 2: Commands for Robot Demonstration in Indoor Environment #1

| Navigational Command | Lang2LTL Result | Code as Policies Result |
|---|---|---|
| 1. go to brown bookshelf, metal desk, wooden desk, kitchen counter, and the blue couch in any order | success | success |
| 2. move to grey door, then bookrack, then brown desk, then counter, then white desk | success | success |
| 3. visit brown wooden desk but only after bookshelf | success | misunderstand the task |
| 4. go from brown bookshelf to white metal desk and only visit each landmark one time | success | misunderstand the task |
| 5. go to brown wooden desk exactly once and do not visit brown desk before bookshelf | success | inexecutable |
| 6. go to white desk at least three times | success | inexecutable |
| 7. go to wooden bookshelf at least five times | success | success |
| 8. visit bookshelf at most three times | success | success |
| 9. visit counter at most 5 times | success | success |
| 10. go to wooden desk exactly three times | success | misunderstand the task |
| 11. move to brown wooden desk exactly 5 times | success | inexecutable |
| 12. go to doorway exactly two times, in addition always avoid the table | success | success |
| 13. go to brown desk only after visiting bookshelf, in addition go to brown desk only after visiting white desk | success | misunderstand the task |
| 14. visit wooden desk exactly two times, in addition do not go to wooden desk before bookrack | success | inexecutable |
| 15. visit wooden desk at least two times, in addition do not go to wooden desk before bookshelf | success | inexecutable |
| 16. visit the blue IKEA couch, in addition never go to the big steel door | success | success |
| 17. visit white kitchen counter then go to brown desk, in addition never visit white table | success | success |
| 18. go to the grey door, and only then go to the bookshelf, in addition always avoid the table | success | misunderstand the task |
| 19. go to kitchen counter then wooden desk, in addition after going to counter, you must avoid white table | success | misunderstand the task |
| 20. Go to bookshelf, alternatively go to metal desk | success | misunderstand the task |
| 21. Go to counter, alternatively go to metal desk | success | misunderstand the task |
| 22. Go to the counter, but never visit the counter | unsatisfiable. abort correctly | stop execution correctly |
| 23. do not go to the wooden desk until bookshelf, and do not go to bookshelf until wooden desk | unsatisfiable. abort correctly | stop execution correctly |
| 24. go to brown desk exactly once, in addition go to brown desk at least twice | unsatisfiable. abort correctly | misunderstand the task |
| 25. find the kitchen counter, in addition avoid the doorway | unsatisfiable. abort correctly | stop execution correctly |
| 26. move to couch exactly twice, in addition pass by counter at most once | unsatisfiable. abort correctly | stop execution correctly |
| 27. navigate to the counter then the brown desk, in addition after going to the counter, you must avoid doorway | unsatisfiable. abort correctly | misunderstand the task |
| 28. Visit the counter at least 2 times and at most 5 times | incorrect grounding. OOD | inexecutable |
| 29. visit counter at least six times | incorrect grounding. OOD | success |
| 30. either go to bookshelf then desk A, or go to couch | incorrect grounding. OOD | misunderstand the task |

| Navigational Command | Lang2LTL Result | Code as Policies Result |
| --- | --- | --- |
| 1. navigate to the office with the wooden door, the classroom with glass door and the table in the atrium, kitchen counter, and the blue couch in any order | success | success |
| 2. go down the hallway decorated with paintings, then find the kitchen table, then front desk, then staircase | success | success |
| 3. navigate to classroom but do not visit classroom before the white table in atrium | success | misunderstand the task |
| 4. only visit classroom once, and do not visit classroom until you visit elevator first | success | success |
| 5. Go to the staircase, front desk and the white table in the atrium in that exact order. You are not permitted to revisit any of these locations | success | inexecutable |
| 6. go to the purple elevator at least five times | success | inexecutable |
| 7. visit the kitchen table at most three times | success | success |
| 8. navigate to the classroom exactly four times | success | inexecutable |
| 9. go to the front desk then the yellow office door, in addition do not visit the classroom with glass door | success | success |
| 10. go to the stairs then the front desk, in addition avoid purple elevator | success | success |
| 11. move to elevator then front desk, in addition avoid staircase | success | success |
| 12. go to front desk exactly two times, in addition avoid elevator | success | inexecutable |
| 13. Go to elevator, alternatively go to staircase | success | misunderstand the task |
| 14. Go to the front desk at least two different occasions, in addition you are only permitted to visit the staircase at most once | success | misunderstand the task |
| 15. Visit the elevator exactly once, in addition visit the front desk on at least 2 separate occasions | success | inexecutable |
| 16. Go to the office, in addition avoid visiting the elevator and the classroom | success | success |
| 17. Visit the front desk, in addition you are not permitted to visit elevator and staircase | success | success |
| 18. Visit the purple door elevator, then go to the front desk and then go to the kitchen table, in addition you can never go to the elevator once you've seen the front desk | success | inexecutable |
| 19. Visit the front desk then the white table, in addition if you visit the staircase you must avoid the elevator after that | success | inexecutable |
| 20. Go to the classroom with glass door, but never visit the classroom with glass door | unsatisfiable. abort correctly | stop execution correctly |
| 21. do not go to the white table until classroom, and do not go to the classroom until white table | unsatisfiable. abort correctly | stop execution correctly |
| 22. go to kitchen table exactly once, in addition go to kitchen table at least twice | unsatisfiable. abort correctly | misunderstand the task |
| 23. find the office, in addition avoid visiting the front desk and the classroom and the table in atrium | unsatisfiable. abort correctly | stop execution correctly |
| 24. move to the kitchen table exactly twice, in addition pass by hallway decorated by paintings at most once | unsatisfiable. abort correctly | misunderstand the task |
| 25. navigate to the kitchen table then the front desk, in addition after going to the kitchen table, you must avoid hallway decorated with paintings | unsatisfiable. abort correctly | misunderstand the task |
| 26. Go to the front desk at least 4 different occasions, additionally, you are only permitted to visit the staircase at most once | incorrect grounding. OOD | inexecutable |
| 27. Visit the front desk, additionally if you visit the elevator you must visit the office after that | incorrect grounding. OOD | success |
| 28. Visit the front desk, additionally you visit the elevator you must visit the office after that the white table and the classroom | incorrect grounding. OOD | misunderstand the task |

Table 4: Specification Patterns for Lang2LTL

| Specification Type | Explanation | Formula |
|---|---|---|
| Visit | Visit a set of waypoints $\{p_1, p_2 \ldots, p_n\}$ in any order | $\bigwedge_{i=1}^{n} \mathbf{F}\, p_i$ |
| Sequence Visit | Visit a set of waypoints $\{p_1, p_2 \ldots, p_n\}$, but ensure that $p_2$ is visited at least once after visiting $p_1$, and so on | $\mathbf{F}(p_1 \wedge \mathbf{F}(p_2 \wedge \ldots \wedge \mathbf{F}(p_n)) \ldots)$ |
| Ordered Visit | Visit a set of waypoints $\{p_1, p_2 \ldots, p_n\}$, but ensure that $p_2$ is never visited before visiting $p_1$ | $\mathbf{F}(p_n) \wedge \bigwedge_{i=1}^{n-1}(\neg p_{i+1}\, \mathbf{U}\, p_i)$ |
| Strictly Ordered Visit | Visit a set of waypoints $\{p_1, p_2 \ldots, p_n\}$, but ensure that $p_2$ is never visited before visiting $p_1$, additionally, ensure that $p_1$ is only visited on a single distinct visit before completing the rest of the task | $\mathbf{F}(p_n) \quad \wedge \quad \bigwedge_{i=1}^{n-1}(\neg p_{i+1}\ \mathbf{U}\ p_i) \ \wedge$ $\bigwedge_{i=1}^{n-1}(\neg p_i\, \mathbf{U}\, (p_i\, \mathbf{U}\, (\neg p_i\, \mathbf{U}\, p_{i+1})))$ |
| Patrolling | Visit a set of wwaypoints $\{p_1, p_2 \ldots, p_n\}$ infinitely often | $\bigwedge_{i=1}^{n} \mathbf{GF}p_i$ |
| Bound Delay | If and only if the proposition $a$ is ever observed, then the proposition $b$ must hold at the very next time step | $\mathbf{G}(a \leftrightarrow \mathbf{X}b)$ |
| Delayed Reaction | If the proposition $a$ is ever observed, then its response is to ensure that the proposition $b$ holds at some point in the future | $\mathbf{G}(a \rightarrow \mathbf{F}b)$ |
| Prompt Reaction | If the proposition $a$ is ever observed, then the proposition $b$ must hold at the very next time step | $\mathbf{G}(a \rightarrow \mathbf{X}b)$ |
| Wait | The proposition $a$ must hold till the proposition $b$ becomes true, and $b$ may never hold | $a\, \mathbf{W}\, b$ |
| Past Avoidance | The proposition $a$ must not become true until the proposition $b$ holds first. $b$ may never hold | $\neg a\, \mathbf{W}\, b$ |
| Future Avoidance | Once the proposition $a$ is observed to be true, the proposition $b$ must never be allowed to become true from that point onwards. | $\mathbf{G}(a \rightarrow \mathbf{XG}\neg b)$ |
| Global Avoidance | The set of propositions $\{p_1, p_2 \ldots, p_n\}$ must never be allowed to become true | $\bigwedge_{i=1}^{n} \mathbf{G}(\neg p_i)$ |
| Upper Restricted Avoidance | The waypoint $a$ can be visited on at most $n$ separate visits | For $n = 1$, $\neg\mathbf{F}(a \wedge (a\, \mathbf{U}\, (\neg a \wedge (\neg a\, U\, \mathbf{F}a))))$ For $n = 2$, $\neg\mathbf{F}(a \wedge (a\, U\, (a \wedge (\neg a\, \mathbf{U}\, \mathbf{F}(a \wedge (a\, \mathbf{U}\, (\neg a \wedge (\neg a\, \mathbf{U}\, Fa)))))))$ |
| Lower Restricted Avoidance | The waypoint $a$ must be visited on at least $n$ separate visits | For $n = 1$, $\neg\mathbf{F}a$ for $n = 2$, $\mathbf{F}(a \wedge (a\, \mathbf{U}\, (\neg a \wedge (\neg a\, U\, Fa))))$ |
| Exact Restricted Avoidance | The waypoint $a$ must be visited on exactly $n$ separate visits | For $n = 1$, $a\, \mathbf{M}\, (\neg a \vee \mathbf{G}(a \vee \mathbf{G}\neg a))$ For $n = 2$, $(a \wedge \mathbf{F}(\neg a \wedge Fa))\mathbf{M}(\neg a \vee \mathbf{G}(a \vee \mathbf{G}(\neg a \vee \mathbf{G}(a \vee \mathbf{G}\neg a))))$ |

Table 5: Example Commands from OpenStreetMap Dataset

| LTL Type | Command (with two referring expressions) |
|---|---|
| Visit | move to Thai hot pot restaurant on Kneeland Street, and Vietnamese restaurant on Washington Street |
| Sequence Visit | visit Subway sandwich shop on The Plaza, followed by Zada Jane's Cafe on Central Avenue |
| Ordered Visit | find Local Goods Chicago gift shop, but not until you find Currency exchange bureau, first |
| Strictly Ordered Visit | reach Citibank branch, and then Cutler Majestic Theater on Tremont Street, in that exact order without repetitions |
| Patrolling | keep on visiting US Post Office on West Devon Avenue, and Kanellos Shoe Repair shop |
| Bound Delay | you must go to Purple Lot parking area, immediately after you visit Royal Nails & Spa on South Main Street, and you can not go to Purple Lot parking area, any other time |
| Delayed Reaction | you must visit Peruvian restaurant on Virginia Avenue, once you visit PNC Bank |
| Prompt Reaction | immediately after you go to Beachside Resortwear clothing store, you must go to Walgreens Pharmacy |
| Wait | you can not go to other place from Publix supermarket, unless you see Beaches Museum |
| Past Avoidance | avoid visiting IES Test Prep school, till you observe bookstore on Elizabeth Street |
| Future Avoidance | never go to Commercial building on 5th Avenue, once you go to Cafe Metro |
| Global Avoidance | make sure to never reach either Citibank, or Seybolt Park |
| Upper Restricted Avoidance | go to Cocktail bar, at most twice |
| Lower Restricted Avoidance | you have to visit Main Branch of CoGo Bike Share Library for bicycle rental, two or more than two times |
| Exact Restricted Avoidance | navigate to Art shop on Bannock Street, exactly twice |

Table 6: Results of Recognizing Referring Expression with Spatial Relations

| Navigational Command | Referring Expression(s) | Correctness |
|---|---|---|
| 1. go to back of Common Market | back of Common Market | correct |
| 2. always avoid entrance and exit of Little Sugar Creek, but visit left and right of Little Sugar Creek | entrance and exit of Little Sugar Creek \| left and right of Little Sugar Creek | correct |
| 3. stay at intersection of Thayer street and Waterman street | intersection of Thayer street and Waterman street | correct |
| 4. move forward to the south of Edgebrook Coffee Ship | south of Edgebrook Coffee Ship | correct |
| 5. go to east of Chinatown, without visiting west of New Saigon Sandwich, then go to front of New Saigon Sandwich, without visiting rear of Dumpling Cafe, then go to rear of Dumpling Cafe, without visiting north of Emerson College - Little Building, finally go to south Emerson College - Little Building, while only visiting each location once | east of Chinatown \| west of New Saigon Sandwich \| front of New Saigon Sandwich — rear of Dumpling Cafe \| rear of Dumpling Cafe \| north of Emerson College - Little Building \| south Emerson College - Little Building | correct |
| 6. go around big blue box | big blue box | incorrect |
| 7. go to exit of blue area through between red room and blue one | exit of blue area \| red room \| blue one | incorrect |
| 8. go to left of CVS and the stay on bridge | left of CVS \| bridge | incorrect |
| 9. go pass right of Dairy Queen to left of Harris Teeter, end up at entrance of Wells Fargo | Dairy Queen \| Harris Teeter \| Wells Fargo | incorrect |
| 10. move to My Sister's Closet and stop close to bus stop near Ace Hardware | My Sister's Closet \| bus stop near Ace Hardware | incorrect |

