# OpenReview forum: "Grounding Complex Natural Language Commands for Temporal Tasks in Unseen Environments"
_robot-learning.org/CoRL/2023/Conference — CoRL 2023 Poster_

### Official Review · Reviewer_qzh3 · 2023-07-17

**Confidence:** 3
**Originality:** Very Good
**Technical Quality:** Very Good
**Clarity Of Presentation:** Excellent
**Impact:** 4

**Recommendation:**

Strong Accept: I recommend accepting the paper and will argue for my recommendation even if other reviewers hold a different opinion.

**Review:**

Strengths
- The problem is well motivated and the approach is well justified.
- Extensive discussion on generalization in understanding temporal navigation commands.
- The claims are generally well supported by experimental results across multiple datasets and task types.
- The paper is clearly written and well structured. The problem setup and proposed approach are easy to understand.
- Detailed experiments across different LLMs.
- Experiments used k-fold validation to report standard deviations
- The limitations section acknowledges overfitting by the lifted translation model and the inability to distinguish between landmarks with the same features. The authors suggest using larger models and dialog for disambiguation.

Weaknesses / Feedback
1) Including some example commands used in the actual experiments would make the experiment clearer. In addition, the authors should provide an analysis of how the recognition and grounding modules perform relative to command/description complexity.
2) A more detailed failure mode analysis for the real robot experiments would make the paper stronger. What proportion of translation errors is due to failures in the grounding vs. lifted translation modules?
3) Are the symbolic formulae unique in LTL notation? If not, what happens when the symbolic translation LLM outputs a different but functionally identical LTL formula?
4) It would be beneficial, from a user experience standpoint, to have an additional LLM that translates the final output formula back into natural language, where the user can preview how the planner understood the instruction and give corrective feedback if necessary.
5) I urge the authors to release code and finetuned models so that the method can be easily reproduced by other researchers.


**Quality Of The Limitations Section:**

Limitations are addressed clearly

**Questions For Rebuttal:**

See feedback section.

**Robotics Focus:**

Sufficient demonstration on hardware

**Summary Of Paper:**

This paper tackles the problem of grounding complex natural language commands into formal linear temporal logic (LTL) specifications for robotic navigation tasks. This is important because LTL provides unambiguous semantics for reasoning about long-horizon tasks and verifying the satisfaction of temporal constraints given in the instruction. Existing approaches require training data from specific environments and landmarks, limiting generalization. The authors propose Lang2LTL, a modular system using LLMs to ground commands to LTL without prior language data from the target environment. Generalization experiments evaluated Lang2LTL on 5 generalization criteria on a new dataset with 2125 unique LTL formulas corresponding to 47 task templates. In system-level evaluations, Lang2LTL outperformed the baseline Code as Policies approach, which does not leverage LTL.

**Summary Of Recommendation:**

The paper is well-motivated and clearly written. The proposed method is a great example of connecting the open-world reasoning capabilities of LLMs with more precise task representations and planners. I recommend acceptance as the paper has convincing experiments, and the method should be easily adoptable by others in the field.

---

### Official Review · Reviewer_y8dy · 2023-07-19

**Confidence:** 3
**Originality:** Good
**Technical Quality:** Good
**Clarity Of Presentation:** Good
**Impact:** 4

**Recommendation:**

Weak Accept: I recommend accepting the paper, but will not argue for my recommendation if the majority of other reviewers have a different opinion.

**Review:**

The explanation of the proposed system is easy to follow. However the evaluation is less clear. In particular, the authors claim that "Lang2LTL achieved state-of-the-art performance in grounding diverse temporal commands in 21 novel OpenStreetMap regions" but I have hard time finding evidence for this in the paper. The only prior work shown in Fig. 3 is the CopyNet model which has 0 accuracy and hence cannot be considered a reasonable baseline. In Table 2, the accuracy of Lang2LTL is either not significantly larger than prior accuracy (in the statistical sense), or is worse than prior accuracy. I'm happy to improve my score if the authors clarify the comparison to baselines (I do appreciate the comparison to code-as-policies).

**Quality Of The Limitations Section:**

Limitations are addressed clearly

**Questions For Rebuttal:**

See the "Review" section.

**Robotics Focus:**

Sufficient demonstration on hardware

**Summary Of Paper:**

The paper presents a system based on LLMs for translating free form language navigation instructions into a structured command (based on linear temporal logic) that can be interpreted by off-the-shelf planners. The paper claims the method is state-of-the-art on this task.

**Summary Of Recommendation:**

Evaluation needs to be more thorough or better documented.

---

### Official Review · Reviewer_UUsu · 2023-07-20

**Confidence:** 4
**Originality:** Very Good
**Technical Quality:** Excellent
**Clarity Of Presentation:** Excellent
**Impact:** 4

**Recommendation:**

Strong Accept: I recommend accepting the paper and will argue for my recommendation even if other reviewers hold a different opinion.

**Review:**

**Strengths**
1. Clarity: The paper's language is lucid, brief, and captivating, ensuring that readers can grasp the concepts effortlessly. The information flows logically, maintaining cohesion throughout the document.
2. Originality: The work is very original, and the proposed approach is novel with promising results.
3. Real-robot results: The paper also showcases results on a physical robot that successfully follows 52 semantically diverse navigational commands in two indoor environments using Lang2LTL.
4. A reasonably complete comparative study of Lang2LTL performance with strong state-of-the-art baselines like Code-as-policies (CaP), CopyNet-based translation model, Prompt GPT-4, etc.
5. Well-organized and informative appendix giving details on data collection, implementation details, and relevant examples as and when needed.
6. Promising results supported by quantitive and qualitative analysis of the proposed system.

**Weaknesses / Issues**

1. Line 136 of the paper (section 5.1) mentions that the currently proposed method is limited to processing language instructions containing noun phrases and proper names. However, it cannot effectively handle indirect references and pronoun phrases like "Go to the post office. Go to 'it' via the tall bridge." This limitation of Lang2LTL is not explicitly addressed elsewhere in the paper.
2. In line 242, what is the Table “6.4” referenced here?
3. In Figure 2, the yellow box for “Symbolic Formula” has an error - It should be “F a & ! a U b” instead of “F b & !a U b”.
4. Line 261 has a typo - “crown-sourced”
5. Line 295 - Appendix Table 1 and Table 2 should be Tables 2 and 3 instead.


**Quality Of The Limitations Section:**

Additional details required

**Questions For Rebuttal:**

1. Please clarify the first point of the weaknesses above. Can Lang2LTL handle pronouns and indirect references to landmarks? If not, can you propose a high-level solution that could be a direction for future work to make this approach more robust?
2. Kindly recheck the above-pointed typos and referencing errors and address them.


**Robotics Focus:**

Sufficient demonstration on hardware

**Summary Of Paper:**

The paper introduces Lang2LTL, a modular system and software package that utilizes large language models (LLMs) to ground navigational commands to linear temporal logic (LTL) specifications in novel environments without requiring retraining, as long as it is provided with a semantic map. Lang2LTL addresses three distinct tasks: referring expression recognition, grounding expressions to real-world landmarks, and translating the lifted command to obtain a grounded linear temporal logic (LTL) specification. This approach demonstrates state-of-the-art performance in grounding navigational commands to diverse temporal specifications in 21 city-scaled environments. The system achieves an 81.83% grounding accuracy in unseen cities, outperforming previous state-of-the-art methods and an end-to-end prompt GPT-4 baseline. The paper also showcases a physical robot successfully following 52 semantically diverse navigational commands in two indoor environments using Lang2LTL.

**Summary Of Recommendation:**

The paper presents a novel and significant contribution to the field of natural language understanding for robotic navigation tasks. By proposing Lang2LTL, the paper addresses the challenges of grounding navigational commands to linear temporal logic (LTL) without the need for re-training in novel environments. The comprehensive evaluation of Lang2LTL showcases impressive results in diverse city-scaled environments, demonstrating its state-of-the-art ability as Lang2LTL outperforms previous state-of-the-art methods, including an end-to-end prompt GPT-4 baseline. The successful demonstration of a physical robot following 52 semantically diverse navigational commands further validates the practicality and effectiveness of the proposed approach. The paper deserves acceptance given its novelty, significant contributions, and impressive results.

---

### Official Review · Reviewer_o4yc · 2023-07-23

**Confidence:** 4
**Originality:** Fair
**Technical Quality:** Fair
**Clarity Of Presentation:** Fair
**Impact:** 2

**Recommendation:**

Weak Reject: I recommend rejecting the paper, but will not argue for my recommendation if the majority of other reviewers have a different opinion.

**Review:**

This work tackles multiple hardware platforms (unlike prior work), it also investigates the role of several LMs both closed models accessed via prompting and those that can be fine-tuned. I believe there are likely several important contributions in this work but it can be difficult to tease them apart and verify next to prior work.  The crux of my concerns are around the definition of diversity and clarity of presentation.

**Quality Of The Limitations Section:**

Additional details required

**Questions For Rebuttal:**

1. Am I correct in understanding that referring expression here is not the traditional notion in computer vision or manipulation that includes spatial references like "left of" but instead only prepositions like "on ___ street"?
2. How were the 10 operators, propositions, and templates chosen? 47 templates feels surprisingly small given size and richness of the models being deployed? While most current work does not use LTL which brings it's own benefits and restrictions, this feels like it might be a very constrained semantic space?
3. Related question about the definition of diversity.  Diverse is used to describe the work several times (e.g. "semantically diverse language") but comparisons to exist work on navigation, for example by reading the appendix, leave me worried that the language and object category space is much smaller than traditional navigation datasets instead most of the diversity being due to business names which would reduce the problem to one of pure linguistic KB knowledge.
4. General request for more explanation of why which models are used where.  The intro/conclusions focus on GPT-4, it's unclear if that's the right takeaway from Fig 3a or how the fine-tuned models would do if used in 3b.  Discussing why GPT-4 is the best choice would be appreciated here.

**Robotics Focus:**

Sufficient demonstration on hardware

**Summary Of Paper:**

The aim of this work is to translate natural language commands for navigation into LTL for execution on multiple platforms.  In this case a quadcopter and the Boston Dynamics Spot. The work extends Berg 2020 by "leveraging LLMs and training and evaluating on a semantically diverse dataset".  The key insights include the factorization of the problem for handling referring expressions versus grounding.  Lifted representations allow for a more general template to be learned where command structure is separated from arguments.

**Summary Of Recommendation:**

The work explores different language models and extends prior work on OSM to also include the Spot robot.  However, it's not clear that the evaluation/domain is truly diverse, the learning problem novel, nor the insights broader than another application of "larger LM is better at symbol manipulation"

---

### Author Response · Authors · 2023-08-08
**Uploaded Revised Paper and Appendix**

We revised and uploaded the paper and the appendix. Revisions based on reviewers' feedback are in blue.

We thank the reviewers for their insightful comments and questions.

---

### Decision · Program_Chairs · 2023-08-30

**Decision:**

Accept (Poster)

**Comment:**

The authors present an approach for grounding navigation commands into linear temporal logic via an LLM. They evaluate this system in several domains, over various behaviors, and with different language model approaches.

Though the reviews were not fully in concensus, primarily as the result is not too surprising, they find that the work is an interesting and thorough investigation of how language models can translate into executable LTL instructions, and thus I recommend acceptance.

Though likely out of scope, the paper could be further improved by showing LTL constraints beyond the navigation domain. Some of the results in the appendix could also be moved to the main body to highlight the breadth of results and Figure 1’s clarity could be improved, e.g., by detailing the method and LTL constraints.